# ADVERSARIAL POLICIES: ATTACKING DEEP REINFORCEMENT LEARNING

**Adam Gleave**[1]   **Michael Dennis**   **Cody Wild**   **Neel Kant**   **Sergey Levine**   **Stuart Russell**
University of California, Berkeley

## ABSTRACT

Deep reinforcement learning (RL) policies are known to be vulnerable to adversarial perturbations to their observations, similar to adversarial examples for classifiers. However, an attacker is not usually able to directly modify another agent's observations. This might lead one to wonder: is it possible to attack an RL agent simply by choosing an *adversarial policy* acting in a multi-agent environment so as to create *natural* observations that are adversarial? We demonstrate the existence of adversarial policies in zero-sum games between simulated humanoid robots with proprioceptive observations, against state-of-the-art victims trained via self-play to be robust to opponents. The adversarial policies reliably win against the victims but generate seemingly random and uncoordinated behavior. We find that these policies are more successful in high-dimensional environments, and induce substantially different activations in the victim policy network than when the victim plays against a normal opponent. Fine-tuning protects a victim against a specific adversary, but the attack method can be successfully reapplied to find a new adversarial policy. Videos are available at `https://adversarialpolicies.github.io/`.

## 1 INTRODUCTION

The discovery of adversarial examples for image classifiers prompted a new field of research into adversarial attacks and defenses (Szegedy et al., 2014). Recent work has shown that deep RL policies are also vulnerable to adversarial perturbations of image observations (Huang et al., 2017; Kos and Song, 2017). However, real-world RL agents inhabit natural environments populated by other agents, including humans, who can only modify another agent's observations via their actions. We explore whether it's possible to attack a victim policy by building an *adversarial policy* that takes actions in a shared environment, inducing *natural* observations which have adversarial effects on the victim.

RL has been applied in settings as varied as autonomous driving (Dosovitskiy et al., 2017), negotiation (Lewis et al., 2017) and automated trading (Noonan, 2017). In domains such as these, an attacker cannot usually directly modify the victim policy's input. For example, in autonomous driving pedestrians and other drivers can take actions in the world that affect the camera image, but only in a physically realistic fashion. They cannot add noise to arbitrary pixels, or make a building disappear. Similarly, in financial trading an attacker can send orders to an exchange which will appear in the victim's market data feed, but the attacker cannot modify observations of a third party's orders.

As a proof of concept, we show the existence of adversarial policies in zero-sum simulated robotics games with proprioceptive observations (Bansal et al., 2018a). The state-of-the-art victim policies were trained via self-play to be robust to opponents. We train each adversarial policy using model-free RL against a fixed black-box victim. We find the adversarial policies reliably beat their victim, despite training for less than 3% of the timesteps initially used to train the victim policies.

Critically, we find the adversaries win by creating natural observations that are adversarial, and not by becoming generally strong opponents. Qualitatively, the adversaries fall to the ground in contorted positions, as illustrated in Figure 1, rather than learning to run, kick or block like normal opponents. This strategy does not work when the victim is 'masked' and cannot see the adversary's position, suggesting that the adversary succeeds by manipulating a victim's observations through its actions.

Having observed these results, we wanted to understand the sensitivity of the attack to the dimensionality of the victim's observations. We find that victim policies in higher-dimensional Humanoid

---

[1]Corresponding author. E-mail: `gleave@berkeley.edu`.

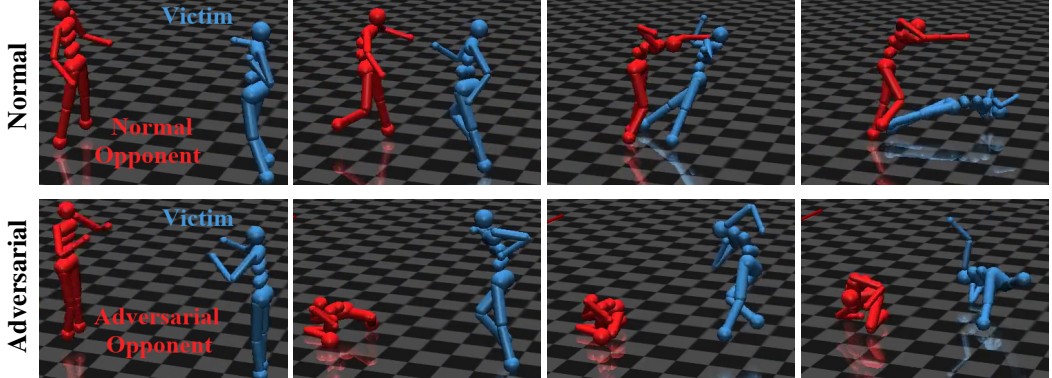

Figure 1: Illustrative snapshots of a victim (in blue) against normal and adversarial opponents (in red). The victim wins if it crosses the finish line; otherwise, the opponent wins. Despite never standing up, the adversarial opponent wins 86% of episodes, far above the normal opponent's 47% win rate.

environments are substantially more vulnerable to adversarial policies than in lower-dimensional Ant environments. To gain insight into why adversarial policies succeed, we analyze the activations of the victim's policy network using a Gaussian Mixture Model and t-SNE (Maaten and Hinton, 2008). We find adversarial policies induce significantly different activations than normal opponents, and that the adversarial activations are typically more widely dispersed between timesteps than normal activations.

A natural defense is to fine-tune the victim against the adversary. We find this protects against that particular adversary, but that repeating the attack method finds a new adversary the fine-tuned victim is vulnerable to. However, this new adversary differs qualitatively by physically interfering with the victim. This suggests repeated fine-tuning might provide protection against a range of adversaries.

Our paper makes three contributions. First, we propose a novel, physically realistic threat model for adversarial examples in RL. Second, we demonstrate the existence of adversarial policies in this threat model for several simulated robotics games. Our adversarial policies reliably beat the victim, despite training with less than 3% as many timesteps and generating seemingly random behavior. Third, we conduct a detailed analysis of why the adversarial policies work. We show they create natural observations that are adversarial to the victim and push the activations of the victim's policy network off-distribution. Additionally, we find policies are easier to attack in high-dimensional environments.

As deep RL is increasingly deployed in environments with potential adversaries, we believe it is important that practitioners are aware of this previously unrecognized threat model. Moreover, even in benign settings, we believe adversarial policies can be a useful tool for uncovering unexpected policy failure modes. Finally, we are excited by the potential of adversarial training using adversarial policies, which could improve robustness relative to conventional self-play by training against adversaries that exploit weaknesses undiscovered by the distribution of similar opponents present during self-play.

## 2 RELATED WORK

Most study of adversarial examples has focused on small $\ell_p$ norm perturbations to images, which Szegedy et al. (2014) discovered cause a variety of models to confidently misclassify the image, even though the changes are visually imperceptible to a human. Gilmer et al. (2018a) argued that attackers are not limited to small perturbations, and can instead construct new images or search for naturally misclassified images. Similarly, Uesato et al. (2018) argue that the near-ubiquitous $\ell_p$ model is merely a convenient local approximation for the true worst-case risk. We follow Goodfellow et al. (2017) in viewing adversarial examples more broadly, as "inputs to machine learning models that an attacker has intentionally designed to cause the model to make a mistake."

The little prior work studying adversarial examples in RL has assumed an $\ell_p$-norm threat model. Huang et al. (2017) and Kos and Song (2017) showed that deep RL policies are vulnerable to small perturbations in image observations. Recent work by Lin et al. (2017) generates a sequence of

perturbations guiding the victim to a target state. Our work differs from these previous approaches by using a physically realistic threat model that disallows direct modification of the victim's observations.

Lanctot et al. (2017) showed agents may become tightly coupled to the agents they were trained with. Like adversarial policies, this results in seemingly strong polices failing against new opponents. However, the victims we attack win against a range of opponents, and so are not coupled in this way.

Adversarial training is a common defense to adversarial examples, achieving state-of-the-art robustness in image classification (Xie et al., 2019). Prior work has also applied adversarial training to improve the robustness of deep RL policies, where the adversary exerts a force vector on the victim or varies dynamics parameters such as friction (Pinto et al., 2017; Mandlekar et al., 2017; Pattanaik et al., 2018). Our defense of fine-tuning the victim against the adversary is inspired by this work.

This work follows a rich tradition of worst-case analysis in RL. In robust MDPs, the transition function is chosen adversarially from an *uncertainty set* (Bagnell et al., 2001; Tamar et al., 2014). Doyle et al. (1996) solve the converse problem: finding the set of transition functions for which a policy is optimal. Methods also exist to verify controllers or find a counterexample to a specification. Bastani et al. (2018) verify decision trees distilled from RL policies, while Ghosh et al. (2018) test black-box closed-loop simulations. Ravanbakhsh et al (2016) can even synthesize controllers robust to adversarial disturbances. Unfortunately, these techniques are only practical in simple environments with low-dimensional adversarial disturbances. By contrast, while our method lacks formal guarantees, it can test policies in complex multi-agent tasks and naturally scales with improvements in RL algorithms.

## 3 FRAMEWORK

We model the victim as playing against an opponent in a two-player Markov game (Shapley, 1953). Our threat model assumes the attacker can control the opponent, in which case we call the opponent an adversary. We denote the adversary and victim by subscript $\alpha$ and $\nu$ respectively. The game $M = (S, (A_\alpha, A_\nu), T, (R_\alpha, R_\nu))$ consists of state set $S$, action sets $A_\alpha$ and $A_\nu$, and a joint state transition function $T : S \times A_\alpha \times A_\nu \to \Delta(S)$ where $\Delta(S)$ is a probability distribution on $S$. The reward function $R_i : S \times A_\alpha \times A_\nu \times S \to \mathbb{R}$ for player $i \in \{\alpha, \nu\}$ depends on the current state, next state and both player's actions. Each player wishes to maximize their (discounted) sum of rewards.

The adversary is allowed unlimited black-box access to actions sampled from $\pi_v$, but is not given any white-box information such as weights or activations. We further assume the victim agent follows a fixed stochastic policy $\pi_v$, corresponding to the common case of a pre-trained model deployed with static weights. Note that in safety critical systems, where attacks like these would be most concerning, it is standard practice to validate a model and then freeze it, so as to ensure that the deployed model does not develop any new issues due to retraining. Therefore, a fixed victim is a realistic reflection of what we might see with RL-trained policies in real-world settings, such as with autonomous vehicles.

Since the victim policy $\pi_\nu$ is held fixed, the two-player Markov game $M$ reduces to a single-player MDP $M_\alpha = (S, A_\alpha, T_\alpha, R'_\alpha)$ that the attacker must solve. The state and action space of the adversary are the same as in $M$, while the transition and reward function have the victim policy $\pi_\nu$ embedded:

$$T_\alpha(s, a_\alpha) = T(s, a_\alpha, a_\nu) \qquad \text{and} \qquad R'_\alpha(s, a_\alpha, s') = R_\alpha(s, a_\alpha, a_\nu, s'),$$

where the victim's action is sampled from the stochastic policy $a_\nu \sim \pi_\nu(\cdot \mid s)$. The goal of the attacker is to find an adversarial policy $\pi_\alpha$ maximizing the sum of discounted rewards:

$$\sum_{t=0}^{\infty} \gamma^t R_\alpha(s^{(t)}, a_\alpha^{(t)}, s^{(t+1)}), \quad \text{where } s^{(t+1)} \sim T_\alpha(s^{(t)}, a_\alpha^{(t)}) \text{ and } a_\alpha \sim \pi_\alpha(\cdot \mid s^{(t)}). \tag{1}$$

Note the MDP's dynamics $T_\alpha$ will be unknown even if the Markov game's dynamics $T$ are known since the victim policy $\pi_\nu$ is a black-box. Consequently, the attacker must solve an RL problem.

## 4 FINDING ADVERSARIAL POLICIES

We demonstrate the existence of adversarial policies in zero-sum simulated robotics games. First, we describe how the victim policies were trained and the environments they operate in. Subsequently, we provide details of our attack method in these environments, and describe several baselines. Finally, we present a quantitative and qualitative evaluation of the adversarial policies and baseline opponents.

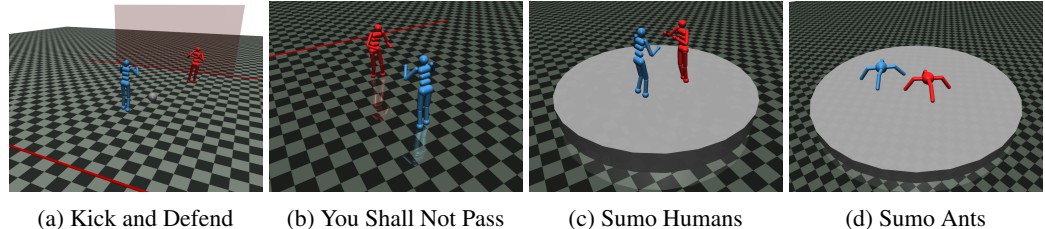

(a) Kick and Defend       (b) You Shall Not Pass       (c) Sumo Humans       (d) Sumo Ants

Figure 2: Illustrations of the zero-sum simulated robotics games from Bansal et al. (2018a) we use for evaluation. Environments are further described in Section 4.1.

## 4.1 Environments and Victim Policies

We attack victim policies for the zero-sum simulated robotics games created by Bansal et al. (2018a), illustrated in Figure 2. The victims were trained in pairs via self-play against random old versions of their opponent, for between 680 and 1360 million timesteps. We use the pre-trained policy weights released in the "agent zoo" of Bansal et al. (2018b). In symmetric environments, the zoo agents are labeled ZooN where $N$ is a random seed. In asymmetric environments, they are labeled ZooVN and ZooON representing the **V**ictim and **O**pponent agents.

All environments are two-player games in the MuJoCo robotics simulator. Both agents observe the position, velocity and contact forces of joints in their body, and the position of their opponent's joints. The episodes end when a win condition is triggered, or after a time limit, in which case the agents draw. We evaluate in all environments from Bansal et al. (2018a) except for *Run to Goal*, which we omit as the setup is identical to *You Shall Not Pass* except for the win condition. We describe the environments below, and specify the number of trained zoo policies and their type (MLP or LSTM):

**Kick and Defend** (3, LSTM). A soccer penalty shootout between two Humanoid robots. The positions of the kicker, goalie and ball are randomly initialized. The kicker wins if the ball goes between the goalposts; otherwise, the goalie wins, provided it remains within 3 units of the goal.

**You Shall Not Pass** (1, MLP). Two Humanoid agents are initialized facing each other. The runner wins if it reaches the finish line; the blocker wins if it does not.

**Sumo Humans** (3, LSTM). Two Humanoid agents compete on a round arena. The players' positions are randomly initialized. A player wins by remaining standing after their opponent has fallen.[2]

**Sumo Ants** (4, LSTM). The same task as *Sumo Humans*, but with 'Ant' quadrupedal robot bodies. We use this task in Section 5.2 to investigate the importance of dimensionality to this attack method.

## 4.2 Methods Evaluated

Following the RL formulation in Section 3, we train an adversarial policy to maximize Equation 1 using Proximal Policy Optimization (PPO; Schulman et al., 2017). We give a sparse reward at the end of the episode, positive when the adversary wins the game and negative when it loses or ties. Bansal et al. (2018a) trained the victim policies using a similar reward, with an additional dense component at the start of training. We train for 20 million timesteps using the PPO implementation from Stable Baselines (Hill et al., 2019). The hyperparameters were selected through a combination of manual tuning and a random search of 100 samples; see Section A in the appendix for details. We compare our methods to three baselines: a policy Rand taking random actions; a lifeless policy Zero that exerts zero control; and all pre-trained policies Zoo* from Bansal et al. (2018a).

## 4.3 Results

**Quantitative Evaluation** We find the adversarial policies reliably win against most victim policies, and outperform the pre-trained Zoo baseline for a majority of environments and victims. We report

---

[2]Bansal et al. (2018a) consider the episode to end in a tie if a player falls before it is touched by an opponent. Our win condition allows for attacks that indirectly modify observations without physical contact.

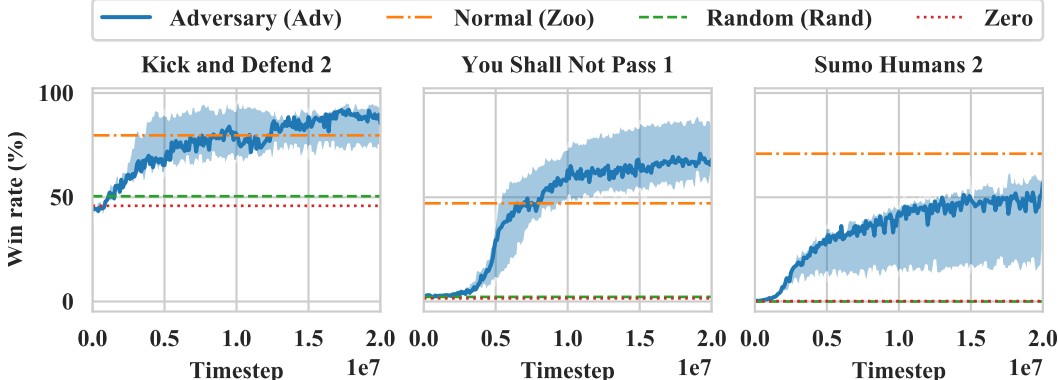

Figure 3: Win rates while training adversary `Adv` against the median victim in each environment (based on the difference between the win rate for `Adv` and `Zoo`). The adversary outperforms the `Zoo` baseline against the median victim in *Kick and Defend* and *You Shall Not Pass*, and is competitive on *Sumo Humans*. For full results, see figure 4 below or figure C.1 in the supplementary material.

**Key**: The solid line shows the median win rate for `Adv` across 5 random seeds, with the shaded region representing the minimum and maximum. The win rate is smoothed with a rolling average over 100 000 timesteps. Baselines are shown as horizontal dashed lines. Agents `Rand` and `Zero` take random and zero actions respectively. The `Zoo` baseline is whichever `ZooM` (*Sumo*) or `ZooOM` (other environments) agent achieves the highest win rate. The victim is `ZooN` (*Sumo*) or `ZooVN` (other environments), where $N$ is given in the title above each figure.

the win rate over time against the median victim in each environment in Figure 3, with full results in Figure C.1 in the supplementary material. Win rates against all victims are summarized in Figure 4.

**Qualitative Evaluation**  The adversarial policies beat the victim not by performing the intended task (e.g. blocking a goal), but rather by exploiting weaknesses in the victim's policy. This effect is best seen by watching the videos at `https://adversarialpolicies.github.io/`. In *Kick and Defend* and *You Shall Not Pass*, the adversarial policy never stands up. The adversary instead wins by positioning their body to induce adversarial observations that cause the victim's policy to take poor actions. A robust victim could easily win, a result we demonstrate in Section 5.1.

This flavor of attacks is impossible in Sumo Humans, since the adversarial policy immediately loses if it falls over. Faced with this control constraint, the adversarial policy learns a more high-level strategy: it kneels in the center in a stable position. Surprisingly, this is very effective against victim 1, which in 88% of cases falls over attempting to tackle the adversary. However, it proves less effective against victims 2 and 3, achieving only a 62% and 45% win rate, below `Zoo` baselines. We further explore the importance of the number of dimensions the adversary can safely manipulate in Section 5.2.

**Distribution Shift**  One might wonder if the adversarial policies win because they are outside the training distribution of the victim. To test this, we evaluate victims against two simple off-distribution baselines: a random policy `Rand` (green) and a lifeless policy `Zero` (red). These baselines win as often as 30% to 50% in *Kick and Defend*, but less than 1% of the time in *Sumo* and *You Shall Not Pass*. This is well below the performance of our adversarial policies. We conclude that most victim policies are robust to off-distribution observations that are not adversarially optimized.

## 5    UNDERSTANDING ADVERSARIAL POLICIES

In the previous section we demonstrated adversarial policies exist for victims in a range of competitive simulated robotics environments. In this section, we focus on understanding why these policies exist. Specifically, we establish that adversarial policies manipulate the victim through their body position; that victims are more vulnerable to adversarial policies in high-dimensional environments; and that activations of the victim's policy network differ substantially when playing an adversarial opponent.

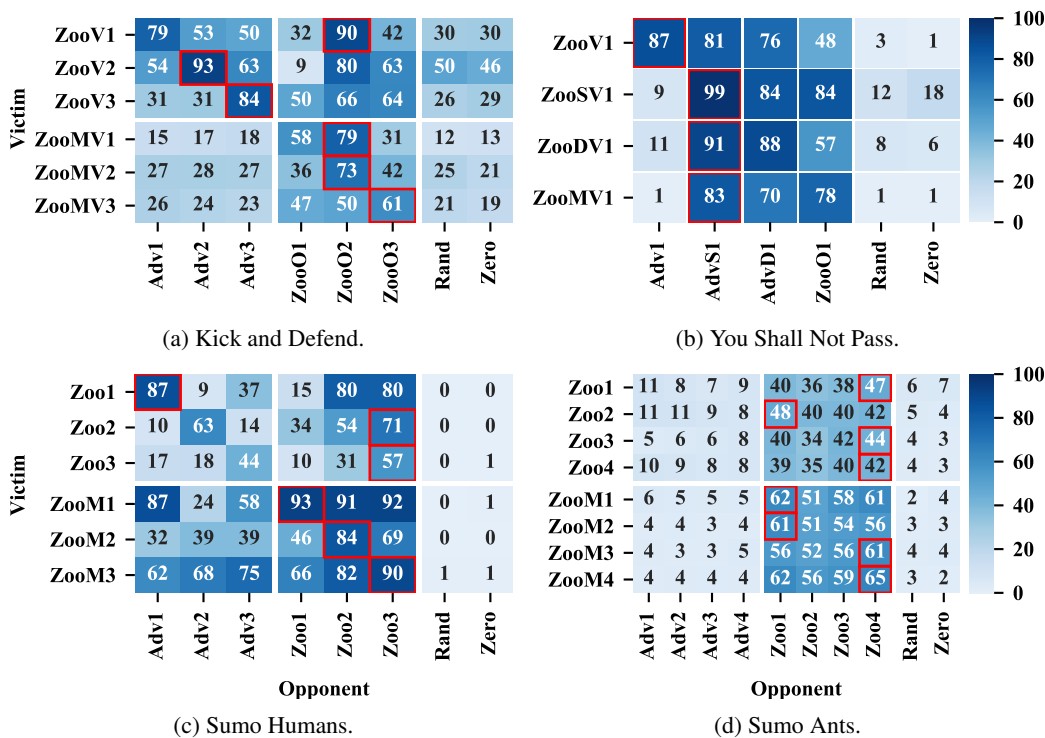

(a) Kick and Defend.

(b) You Shall Not Pass.

(c) Sumo Humans.

(d) Sumo Ants.

Figure 4: Percentage of games won by opponent (out of 1000), the maximal cell in each row is in red. **Key**: Agents `ZooYN` are pre-trained policies from Bansal et al. (2018a), where $Y \in \{'V', 'O', '\,'\}$ denotes the agent plays as **(V)**ictim, **(O)**pponent or either side, and $N$ is a random seed. Opponents `AdvN` are the best adversarial policy of 5 seeds trained against the corresponding `Zoo[V]N`. Agents `Rand` and `Zero` are baseline agents taking random and zero actions respectively. Defended victims `ZooXYN`, where $X \in \{'S', 'D', 'M'\}$, are derived from `ZooYN` by fine-tuning against a **(S)**ingle opponent `AdvN`, **(D)**ual opponents `AdvN` and `Zoo[O]N`, or by **(M)**asking the observations.

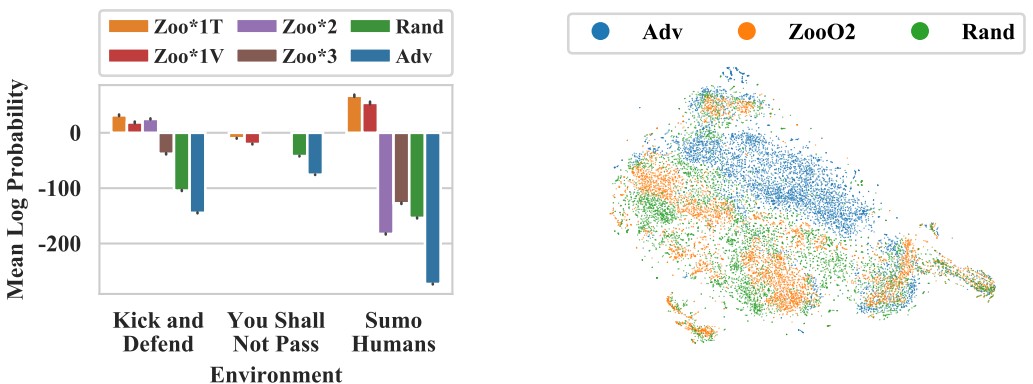

(a) Gaussian Mixture Model (GMM): likelihood the activations of a victim's policy network are "normal". We collect activations for $20,000$ timesteps of victim `Zoo[V]1` playing against each opponent. We fit a 20-component GMM to activations induced by `Zoo[O]1`. Error bars are a 95% confidence interval.

(b) t-SNE activations of Kick and Defend victim `ZooV2` playing against different opponents. Model fitted with a perplexity of 250 to activations from 5000 timesteps against each opponent. See Figures C.3 and C.4 in the supplementary results for visualizations of other environments and victims.

Figure 5: Analysis of activations of the victim's policy network. Both figures show the adversary `Adv` induces off-distribution activations. **Key**: legends specify opponent the victim played against. `Adv` is the best adversary trained against the victim, and `Rand` is a policy taking random actions. `Zoo*N` corresponds to `ZooN` (Sumo) or `ZooON` (otherwise). `Zoo*1T` and `Zoo*1V` are the train and validation datasets, drawn from `Zoo1` (Sumo) or `ZooO1` (otherwise).

## 5.1 MASKED POLICIES

We have previously shown that adversarial policies are able to reliably win against victims. In this section, we demonstrate that they win by taking actions to induce natural observations that are adversarial to the victim, and not by physically interfering with the victim. To test this, we introduce a 'masked' victim (labeled `ZooMN` or `ZooMVN`) that is the same as the normal victim `ZooN` or `ZooVN`, except the observation of the adversary's position is set to a static value corresponding to a typical initial position. We use the same adversarial policy against the normal and masked victim.

One would expect it to be beneficial to be able to see your opponent. Indeed, the masked victims do worse than a normal victim when playing normal opponents. For example, Figure 4b shows that in *You Shall Not Pass* the normal opponent `ZooO1` wins 78% of the time against the masked victim `ZooMV1` but only 47% of the time against the normal victim `ZooV1`. However, the relationship is reversed when playing an adversary. The normal victim `ZooV1` loses 86% of the time to adversary `Adv1` whereas the masked victim `ZooMV1` wins 99% of the time. This pattern is particularly clear in *You Shall Not Pass*, but the trend is similar in other environments.

This result is surprising as it implies highly non-transitive relationships may exist between policies even in games that seem to be transitive. A game is said to be transitive if policies can be ranked such that higher-ranked policies beat lower-ranked policies. Prima facie, the games in this paper seem transitive: professional human soccer players and sumo wrestlers can reliably beat amateurs. Despite this, there is a non-transitive relationship between adversarial policies, victims and masked victims. Consequently, we urge caution when using methods such as self-play that assume transitivity, and would recommend more general methods where practical (Balduzzi et al., 2019; Brown et al., 2019).

Our findings also suggest a trade-off in the size of the observation space. In benign environments, allowing more observation of the environment increases performance. However, this also makes the agent more vulnerable to adversaries. This is in contrast to an idealized Bayesian agent, where the value of information is always non-negative (Good, 1967). In the following section, we investigate further the connection between vulnerability to attack and the size of the observation space.

## 5.2 DIMENSIONALITY

A variety of work has concluded that classifiers are more vulnerable to adversarial examples on high-dimensional inputs (Gilmer et al., 2018b; Khoury and Hadfield-Menell, 2018; Shafahi et al., 2019). We hypothesize a similar result for RL policies: the greater the dimensionality of the component $P$ of the observation space under control of the adversary, the more vulnerable the victim is to attack. We test this hypothesis in the Sumo environment, varying whether the agents are Ants or Humanoids. The results in Figures 4c and 4d support the hypothesis. The adversary has a much lower win-rate in the low-dimensional *Sumo Ants* ($\dim P = 15$) environment than in the higher dimensional *Sumo Humans* ($\dim P = 24$) environment, where $P$ is the position of the adversary's joints.

## 5.3 VICTIM ACTIVATIONS

In Section 5.1 we showed that adversarial policies win by creating natural observations that are adversarial to the victim. In this section, we seek to better understand *why* these observations are adversarial. We record activations from each victim's policy network playing a range of opponents, and analyze these using a Gaussian Mixture Model (GMM) and a t-SNE visualization. See Section B in the supplementary material for details of training and hyperparameters.

We fit a GMM on activations `Zoo*1T` collected playing against a normal opponent, `Zoo1` or `ZooV1`, holding out `Zoo*1V` for validation. Figure 5a shows that the adversarial policy `Adv` induces activations with the lowest log-likelihood, with random baseline `Rand` only slightly more probable. Normal opponents `Zoo*2` and `Zoo*3` induce activations with almost as high likelihood as the validation set `Zoo*1V`, except in *Sumo Humans* where they are as unlikely as `Rand`.

We plot a t-SNE visualization of the activations of Kick and Defend victim `ZooV2` in Figure 5b. As expected from the density model results, there is a clear separation between between `Adv`, `Rand` and the normal opponent `ZooO2`. Intriguingly, `Adv` induces activations more widely dispersed than the random policy `Rand`, which in turn are more widely dispersed than `ZooO2`. We report on the full set of victim policies in Figures C.3 and C.4 in the supplementary material.

## 6 Defending Against Adversarial Policies

The ease with which policies can be attacked highlights the need for effective defenses. A natural defense is to fine-tune the victim zoo policy against an adversary, which we term *single* training. We also investigate *dual* training, randomly picking either an adversary or a zoo policy at the start of each episode. The training procedure is otherwise the same as for adversaries, described in Section 4.2.

We report on the win rates in *You Shall Not Pass* in Figure 4b. We find both the single `ZooSV1` and dual `ZooDV1` fine-tuned victims are robust to adversary `Adv1`, with the adversary win rate dropping from 87% to around 10%. However, `ZooSV1` catastrophically forgets how to play against the normal opponent `ZooO1`. The dual fine-tuned victim `ZooDV1` fares better, with opponent `ZooO1` winning only 57% of the time. However, this is still an increase from `ZooO1`'s 48% win rate against the original victim `ZooV1`. This suggests `ZooV1` may use features that are helpful against a normal opponent but which are easily manipulable (Ilyas et al., 2019).

Although the fine-tuned victims are robust to the original adversarial policy `Adv1`, they are still vulnerable to our attack method. New adversaries `AdvS1` and `AdvD1` trained against `ZooSV1` and `ZooDV1` win at equal or greater rates than before, and transfer successfully to the original victim. However, the new adversaries `AdvS1` and `AdvD1` are qualitatively different, tripping the victim up by lying prone on the ground, whereas `Adv1` causes `ZooV1` to fall without ever touching it.

## 7 Discussion

**Contributions**. Our paper makes three key contributions. **First**, we have proposed a novel threat model of *natural* adversarial observations produced by an adversarial policy taking actions in a shared environment. **Second**, we demonstrate that adversarial policies exist in a range of zero-sum simulated robotics games against state-of-the-art victims trained via self-play to be robust to adversaries. **Third**, we verify the adversarial policies win by confusing the victim, not by learning a generally strong policy. Specifically, we find the adversary induces highly off-distribution activations in the victim, and that victim performance *increases* when it is blind to the adversary's position.

**Self-play**. While it may at first appear unsurprising that a policy trained as an adversary against another RL policy would be able to exploit it, we believe that this observation is highly significant. The policies we have attacked were explicitly trained via self-play to be robust. Although it is known that self-play with deep RL may not converge, or converge only to a local rather than global Nash, self-play has been used with great success in a number of works focused on playing adversarial games directly against humans (Silver et al., 2018; OpenAI, 2018). Our work shows that even apparently strong self-play policies can harbor serious but hard to find failure modes, demonstrating these theoretical limitations are practically relevant and highlighting the need for careful testing.

Our attack provides some amount of testing by constructively lower-bounding the exploitability of a victim policy – its performance against its worst-case opponent – by training an adversary. Since the victim's win rate declines against our adversarial policy, we can confirm that the victim and its self-play opponent were not in a global Nash. Notably we expect our attack to succeed even for policies in a local Nash, as the adversary is trained starting from a random point that is likely outside the victim's attractive basin.

**Defense**. We implemented a simple defense: fine-tuning the victim against the adversary. We find our attack can be successfully reapplied to beat this defense, suggesting adversarial policies are difficult to eliminate. However, the defense does appear to protect against attacks that rely on confusing the victim: the new adversarial policy is forced to instead trip the victim up. We therefore believe that scaling up this defense is a promising direction for future work. In particular, we envisage a variant of population-based training where new agents are continually added to the pool to promote diversity, and agents train against a fixed opponent for a prolonged period of time to avoid local equilibria.

**Conclusion**. Overall, we are excited about the implications the adversarial policy model has for the robustness, security and understanding of deep RL policies. Our results show the existence of a previously unrecognized problem in deep RL, and we hope this work encourages other researchers to investigate this area further. Videos and other supplementary material are available online at `https://adversarialpolicies.github.io/` and our source code is available on GitHub at `https://github.com/HumanCompatibleAI/adversarial-policies`.

ACKNOWLEDGMENTS

We thank Jakob Foerster, Matthew Rahtz, Dylan Hadfield-Menell, Catherine Olsson, Jan Leike, Rohin Shah, Victoria Krakovna, Daniel Filan, Steven Wang, Dawn Song, Sam Toyer and Dan Hendrycks for their suggestion and helpful feedback on earlier drafts of this paper. We thank Chris Northwood for assistance developing the website accompanying this paper. We are also grateful to our anonymous reviewers for valuable feedback and encouragement to explore defenses in this paper.

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

## A    Training: hyperparameters and computational infrastructure

| Parameter | Value | Search Range | Search Distribution |
|---|---|---|---|
| Total Timesteps | $20 \times 10^6$ | $[0, 40 \times 10^6]$ | Manual |
| Batch size | 16 384 | $[2048, 65\,536]$ | Log uniform |
| Number of environments | 8 | $[1, 16]$ | Manual |
| Mini-batches | 4 | $[1, 128]$ | Log uniform |
| Epochs per update | 4 | $[1, 11]$ | Uniform |
| Learning rate | $3 \times 10^{-4}$ | $[1 \times 10^{-5}, 1 \times 10^{-2}]$ | Log uniform |
| Discount | 0.99 | — | — |
| Maximum Gradient Norm | 0.5 | — | — |
| Clip Range | 0.2 | — | — |
| Advantage Estimation Discount | 0.95 | — | — |
| Entropy coefficient | 0.0 | — | — |
| Value Function Loss Coefficient | 0.5 | — | — |

Table A.1: Hyperparameters for Proximal Policy Optimization.

Table A.1 specifies the hyperparameters used for training. The number of environments was chosen for performance reasons after observing diminishing returns from using more than 8 parallel environments. The total timesteps was chosen by inspection after observing diminishing returns to additional training. The batch size, mini-batches, epochs per update, entropy coefficient and learning rate were tuned via a random search with 100 samples on two environments, *Kick and Defend* and *Sumo Humans*. All other hyperparameters are the defaults in the PPO2 implementation in Stable Baselines (Hill et al., 2019).

We repeated the hyperparameter sweep for fine-tuning victim policies for the defense experiments, but obtained similar results. For simplicity, we therefore chose to use the same hyperparameters throughout.

We used a mixture of in-house and cloud infrastructure to perform these experiments. It takes around 8 hours to train an adversary for a single victim using 4 cores of an Intel Xeon Platinum 8000 (Skylake) processor.

## B    Activation Analysis: t-SNE and GMM

We collect activations from all feed forward layers of the victim's policy network. This gives two 64-length vectors, which we concatenate into a single 128-dimension vector for analysis with a Gaussian Mixture Model and a t-SNE representation.

### B.1    t-SNE hyperparameter selection

We fit models with perplexity 5, 10, 20, 50, 75, 100, 250 and 1000. We chose 250 since qualitatively it produced the clearest visualization of data with a moderate number of distinct clusters.

### B.2    Gaussian Mixture Model hyperparameter selection

We fit models with 5, 10, 20, 40 and 80 components with a full (unrestricted) and diagonal covariance matrix. We used the Bayesian Information Criterion (BIC) and average log-likelihood on a held-out validation set as criteria for selecting hyperparameters. We found 20 components with a full covariance matrix achieved the lowest BIC and highest validation log-likelihood in the majority of environment-victim pairs, and was the runner-up in the remainder.

## C    Figures

Supplementary figures are provided on the subsequent pages.

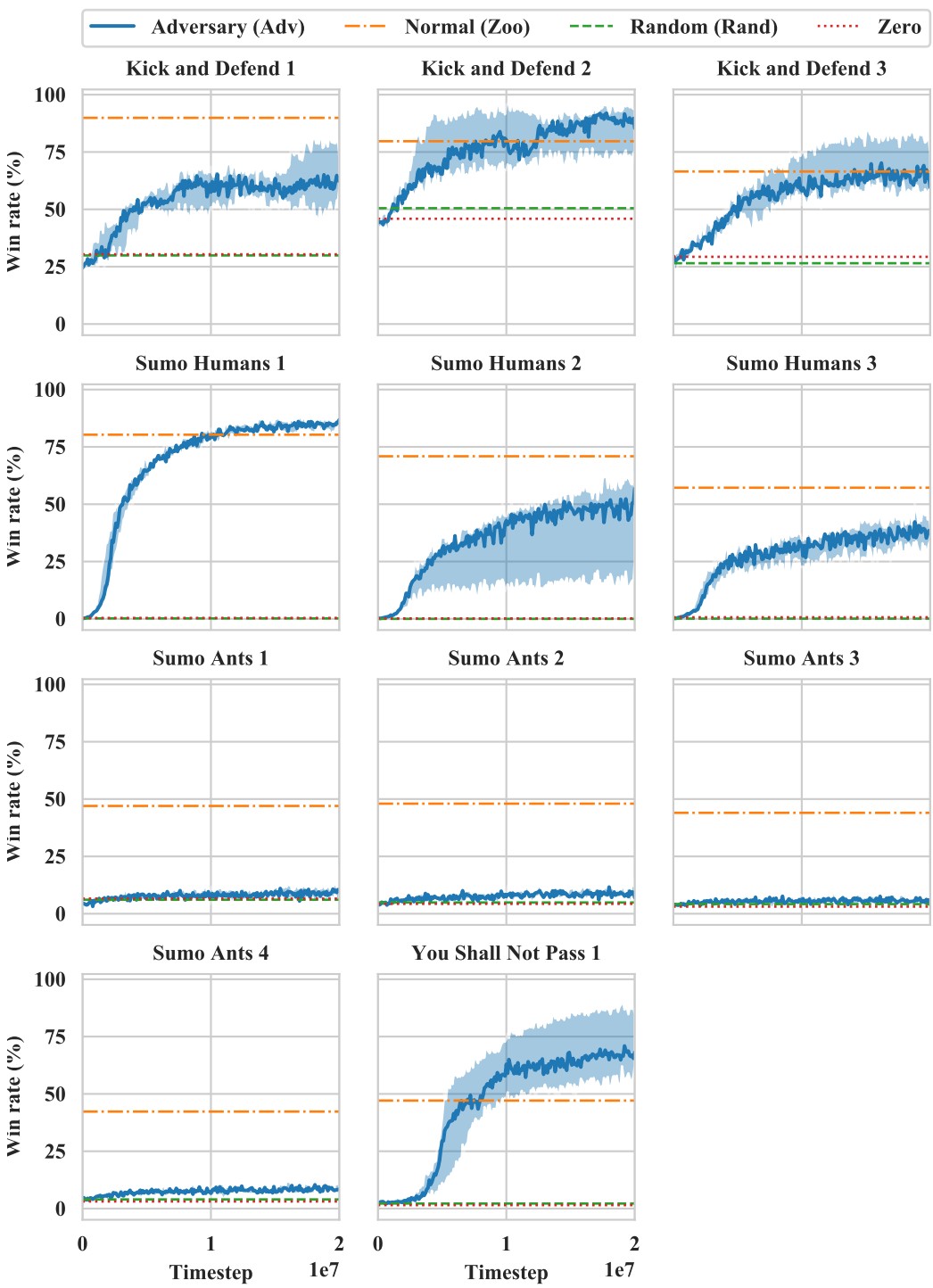

Figure C.1: Win rates while training adversary `Adv`. The adversary exceeds baseline win rates against most victims in *Kick and Defend* and *You Shall Not Pass*, is competitive on *Sumo Humans*, but performs poorly in the low-dimensional *Sumo Ants* environment. **Key**: The solid line shows the median win rate for `Adv` across 5 random seeds, with the shaded region representing the minimum and maximum. The win rate is smoothed with a rolling average over 100 000 timesteps. Baselines are shown as horizontal dashed lines. Agents `Rand` and `Zero` take random and zero actions respectively. The `Zoo` baseline is whichever `ZooM` (*Sumo*) or `ZooOM` (other environments) agent achieves the highest win rate. The victim is `ZooN` (*Sumo*) or `ZooVN` (other environments), where $N$ is given in the title above each figure.

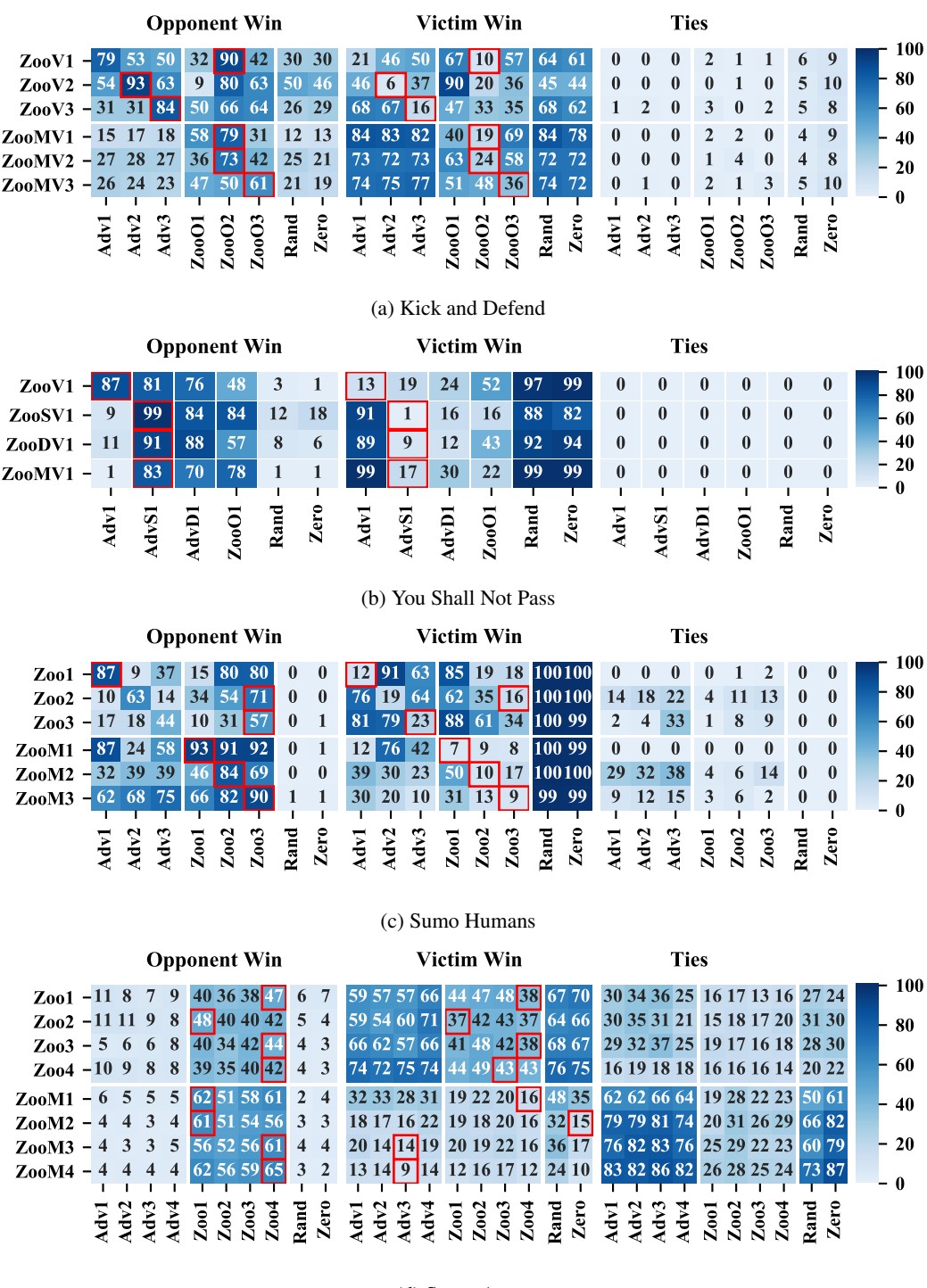

Figure C.2: Percentage of episodes (out of 1000) won by the opponent, the victim or tied. The maximal opponent win rate in each row is in red. Victims are on the $y$-axis and opponents on the $x$-axis. **Key**: Agents `ZooYN` are pre-trained policies from Bansal et al. (2018a), where $Y \in \{`V'`, `O'`, `'`\}$ denotes the agent plays as (**V**)ictim, (**O**)pponent or either side, and $N$ is a random seed. Opponents `AdvN` are the best adversarial policy of 5 seeds trained against the corresponding `Zoo[V]N`. Agents `Rand` and `Zero` are baseline agents taking random and zero actions respectively. Defended victims `ZooXYN`, where $X \in \{`S'`, `D'`, `M'`\}$, are derived from `ZooYN` by fine-tuning against a (**S**)ingle opponent `AdvN`, (**D**)ual opponents `AdvN` and `Zoo[O]N`, or by (**M**)asking the observations.

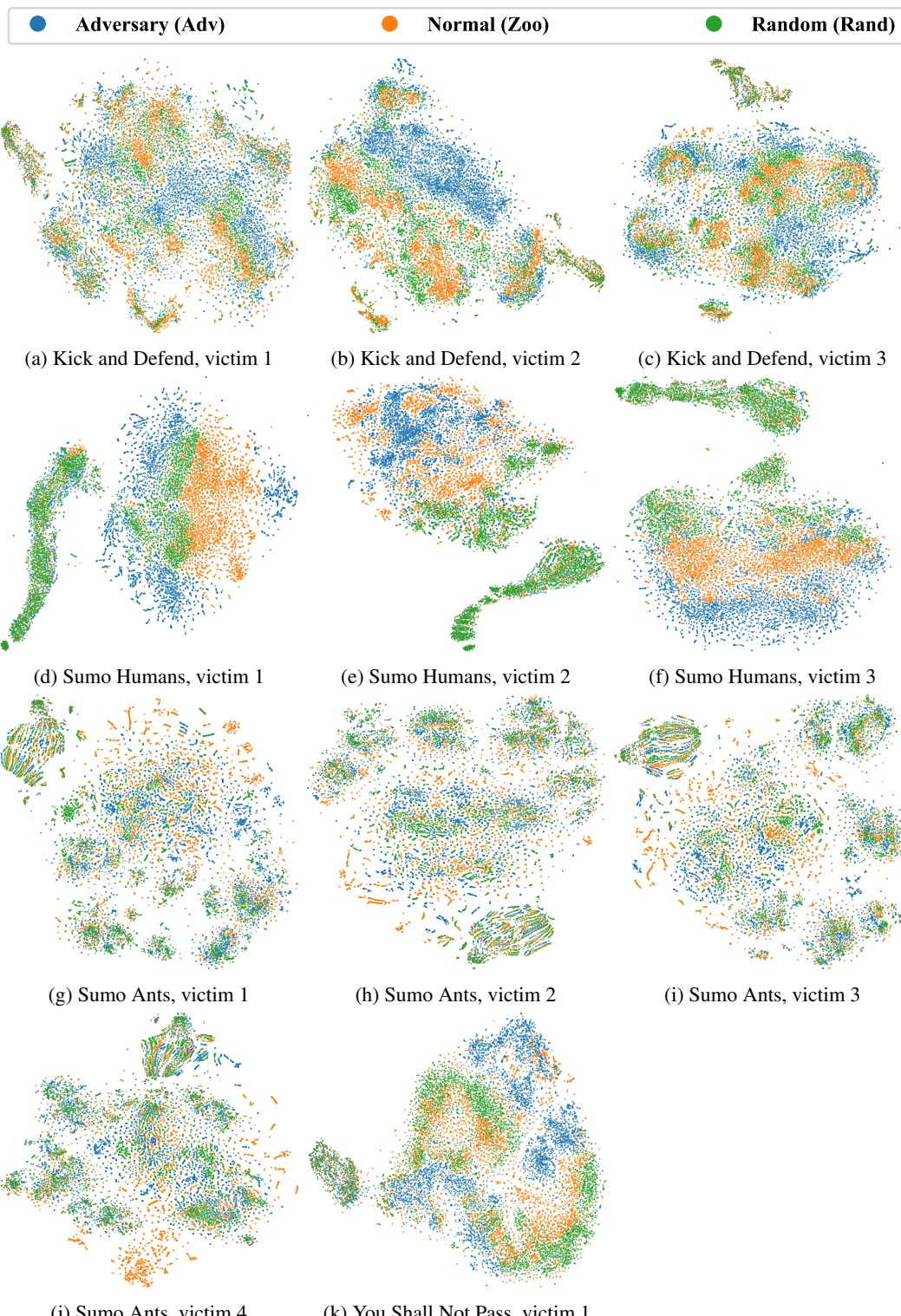

Figure C.3: t-SNE activations of the victim when playing against different opponents. There is a clear separation between the activations induced by `Adv` and those of the normal opponent `Zoo`. Model fitted with a perplexity of 250 to activations from 5000 timesteps against each opponent. The victim is `ZooN` (*Sumo*) or `ZooVN` (other environments), where $N$ is given in the figure caption. Opponent `Adv` is the best adversary trained against the victim. Opponent `Zoo` corresponds to `ZooN` (*Sumo*) or `ZooON` (other environments). See Figure C.4 for activations for a single opponent at a time.

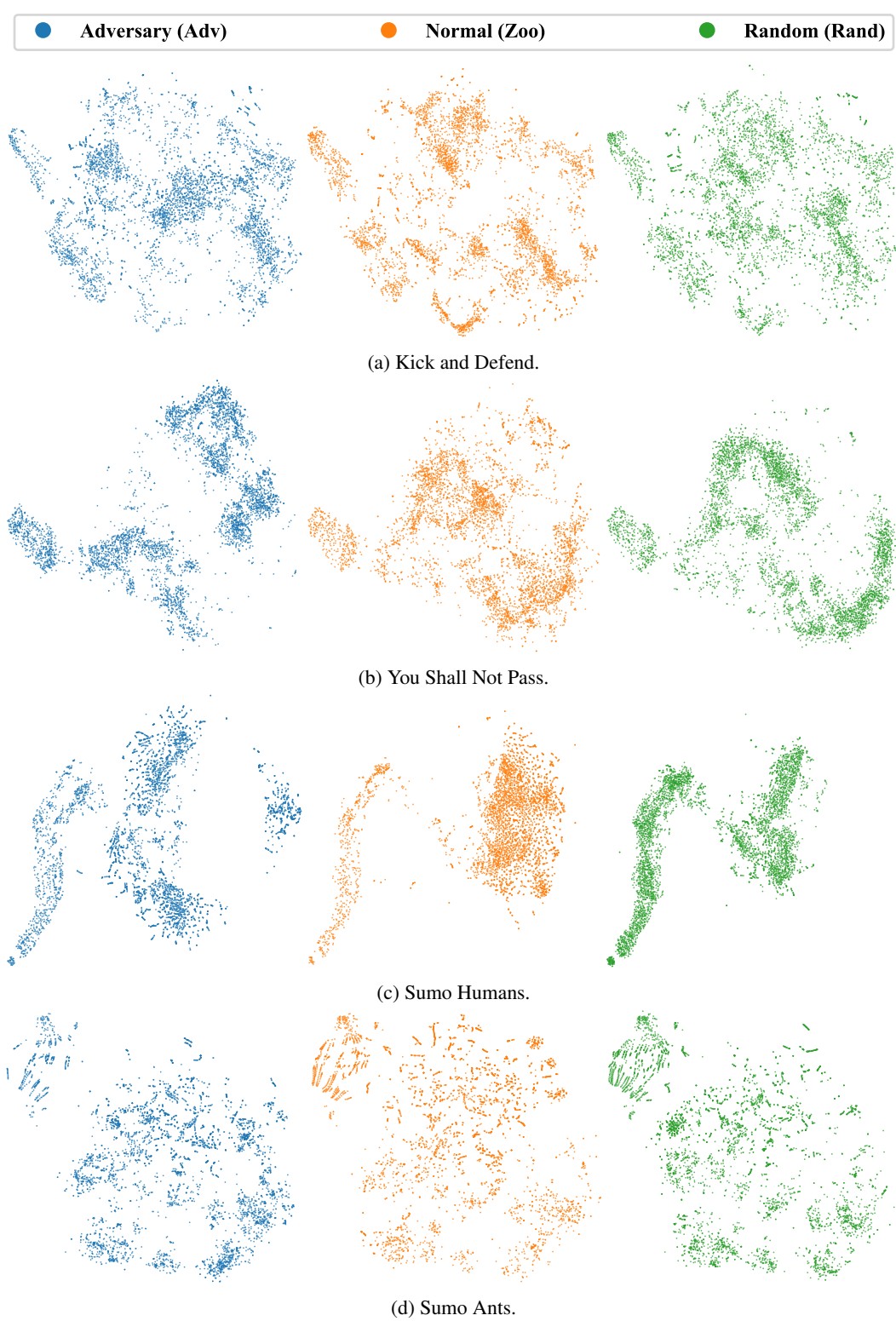

(a) Kick and Defend.

(b) You Shall Not Pass.

(c) Sumo Humans.

(d) Sumo Ants.

Figure C.4: t-SNE activations of victim `Zoo1` (*Sumo*) or `ZooV1` (other environments). The results are the same as in Figure C.3 but decomposed into individual opponents for clarity. Model fitted with a perplexity of 250 to activations from 5000 timesteps against each opponent. Opponent `Adv` is the best adversary trained against the victim. Opponent `Zoo` is `Zoo1` (*Sumo*) or `ZooO1` (other environments). See Figure C.3 for results for other victims (one plot per victim).

