# OpenReview forum: "Adversarial Policies: Attacking Deep Reinforcement Learning"
_ICLR.cc/2020/Conference — Accept (Poster)_

### Official Review · AnonReviewer1 · 2019-10-19
**Official Blind Review #1**

**Rating:** 6

**Review:**

Adversarial Policies: Attacking Deep Reinforcement Learning

In this paper, the authors tackle the problem of adversarial attack by controlling the adversarial policy in reinforcement learning.
I tend to vote for acceptance for this paper, but I also want to point out there is a lot of room for a better experiment section.

Pros:
- In general, the proposed attacking scenarios is novel and missing from the current adversarial attack research, which mostly attack by adding constraint noise in the image space, instead of directly train a new adversarial controlling policy.

- The problem is well formulated and the experiments are also sufficient to support the claim (could be better though, see Cons below).
The visualization provides useful insight for us to deepen understanding of the policy attack.


Cons:

- Experiments limited in sumo cases.
The sumo experiment is quite difficult to train by itself and does not cover the board range of difficulty in multi-agent reinforcement learning tasks.
To be more specific, it will help the generalization of the paper by including
1) Low-dimensional multi-agent environments. It will be even better if the input space is image and the output space is discrete.
2) The study does not consider cooperative multi-agent games. It is also a very important and natural extension to the paper to consider the malicious attack in cooperative games, which can be equally common in real life.

- No solution is given to the problem to combat policy attack.
Some of the results of easy baselines such as iteratively playing attack and defense should be included in the current version of the paper.
It doesn’t have to be successful, but should be quite necessary to increase the research value of the paper.


**Experience Assessment:**

I have published in this field for several years.

**Review Assessment: Checking Correctness Of Derivations And Theory:**

I carefully checked the derivations and theory.

**Review Assessment: Checking Correctness Of Experiments:**

I carefully checked the experiments.

**Review Assessment: Thoroughness In Paper Reading:**

I read the paper thoroughly.

---

> ### Author Response · Authors · 2019-11-09
> **Response**
>
> Thank you for your review and in particular the suggestions for additional experiments. We are glad that you find the attack scenario novel and the problem well formulated. You identify two additional experiments: implementation of defences, and testing on a greater range of environments.
>
> We are working on a defence based on adversarial training, and will revise the paper early next week after further investigating the results. Our preliminary results suggest that the efficacy of the adversarial policy can be reduced by finetuning the victim on a mixture of episodes against the adversary and a normal opponent. (Finetuning against just the adversary works even better, but the victim catastrophically forgets how to play against a normal opponent.) Unfortunately, the victim is hardened only against the original adversary: retraining the adversary recovers comparable performance to the original victim.
>
> Thank you for your suggestion of new types of environment to investigate. We are excited to explore adversarial policies in a broader range of settings, and would appreciate it if you could elaborate on what aspects of the proposed environments you feel is most important.
>
> For property 1), could you clarify what aspect of the environments you think should be low-dimensional?
> In our experiments the agents directly observe the positions and velocities of their and opponents joints, which is low-dimensional compared to image inputs. Our working hypothesis is that vulnerability to adversarial attack depends on the size of the subspace of observations the adversary can control, which depends on the action space, observation space and environment dynamics.
>
> With property 2), we agree that cooperative games are both important and common in real-life. However, we believe that it is harder to attack constant-sum (competitive) games than variable-sum (cooperative) games. In a constant-sum game, the victim agent always plays against a competitor during self-play, and so should learn to be robust to a wide range of strategies. By contrast, in a mixed-sum game, strategies that would reduce the payoffs to all actors will rarely if ever be played by an opponent. This makes cooperative victims easy to exploit by an adversary whose only goal is to minimize the utility of the victim. Accordingly, we feel it is premature to investigate variable-sum games while defense remains unsolved for the constant-sum case.
>
> We would also appreciate any concrete suggestions for environments to investigate. While we are unlikely to have time to perform experiments in new environments during the review period, this is an active direction for our future work. We previously investigated the Carla environment, a driving simulator, which has the benefit of being a safety-critical environment and also has 1) a cooperative element and 2) low-dimensional action spaces. Unfortunately we have found existing controllers in Carla are sufficiently fragile that attacking them is uninformative. We are considering instead attacking agents in simpler top-down car simulators where it may be easier to train a high-performing controller.

---

> > ### Author Response · Authors · 2019-11-14
> > **Defence Update**
> >
> > Thank you again for suggesting considering defences. We have added Section 6 to the paper that describes a defence based on retraining the victim, and a subsequent attack on the victim. We find this defence is effective at preventing the original adversarial policy and similar styles, but that other adversarial policies can still be found using our method.
> >
> > Unfortunately we did not have the time to try iteratively retraining, since it takes around 8 hours to retrain each policy. However, this is a simple extension that we intend to investigate further after the review deadline, and will certainly include in the camera-ready if accepted.
> >
> > We would welcome any additional feedback you may have on defences.

---

> > > ### Author Response · Authors · 2019-11-15
> > > **Defence Update v2**
> > >
> > > We have made our final revision of the paper, updating the intro & related work in light of the defences. We have also updated the website to include videos of the fine-tuned victims and new adversaries, they are accessible under Load Preset->"You Shall Not Pass - Defences" and "You Shall Not Pass - Retrained Adversary".

---

### Official Review · AnonReviewer3 · 2019-10-20
**Official Blind Review #3**

**Rating:** 6

**Review:**

Thank the authors for the response. The update looks good to me. The discussion section looks much better now.
----------------------------------------
Summary
This paper conducts research on adversarial policy against a fixed and black-box policy (victim). In this setting, the victim has a fixed policy but the adversary has no access to its white-box information. Instead, the adversary can freely access the black-box policy of the victim. The experiments show that the trained adversary outperforms the baseline in some scenarios, with three interesting findings: 1) the adversary successfully found the weakness of the victim. In some scenarios, the adversary wins by doing some weird actions which make no sense, but those weird actions somehow make the victim fail; 2) the victim fails due to the weird (or I should say adversarial) observation from the adversary but not the physical inference. The authors demonstrate this by showing that the victim has a much higher success rate when the observation is replaced with a ‘normal’ opponent. I am kind of on the borderline, but still lean to accept paper.
Strengths
- This paper presents some experiments on adversarial learning when the policy of the victim is fixed and black-box with interesting findings, demonstrates that the adversary can successfully figure out the weakness of the victim.
- The paper formalizes the problem into an MDP whose dynamics is unknown, which is clear.
- This paper comes with many experiments supported by great demos, which clearly support the authors’ arguments.
Weaknesses
- The paper becomes messy in the end and does not come with a good conclusion. With a lot of experiments conducted, the author fails to summarize them into an (or a few) interesting conclusion(s), but end up with a page-long conclusion paragraph, which makes the paper less focused but like listing miscellaneous experiments. From the reader’s perspective, a (or a few) clear conclusion would be very helpful.
Possible Improvements
My suggestion is to reorganize the second half of the paper, to make a few clear arguments. Currently, the paper looks pretty narrative, and the author might easily get lost while reading the paper.

**Experience Assessment:**

I have published one or two papers in this area.

**Review Assessment: Checking Correctness Of Derivations And Theory:**

I assessed the sensibility of the derivations and theory.

**Review Assessment: Checking Correctness Of Experiments:**

I assessed the sensibility of the experiments.

**Review Assessment: Thoroughness In Paper Reading:**

I read the paper at least twice and used my best judgement in assessing the paper.

---

> ### Author Response · Authors · 2019-11-09
> **Response**
>
> We’re glad that you found the experimental results interesting and that you agree they clearly support our argument. Your concerns center around the clarity of the second-half of the paper, and especially the Discussion section on the final page.
>
> Thank you for your feedback. In retrospect, the Discussion section was too crowded, trying to present a conclusion, discussion of consequences and future work. We have revised this section to be more tightly focused, and have introduced sub-headings to signpost different material. Two paragraphs justifying our threat model have been removed, with an abbreviated version included in the Framework section. The Contributions sub-section has been rewritten to be itemized and less narrative.
>
> Clarity is important to us, and so we would appreciate your thoughts on the revised paper. We are happy to continue the editing process to improve comprehensibility. It would be particularly helpful to know if there are any specific sections of the paper you find confusing, or if there are any overarching questions about the work you feel the paper did not address.

---

### Official Review · AnonReviewer2 · 2019-10-28
**Official Blind Review #2**

**Rating:** 6

**Review:**

The authors start with the valid observation that for controlled systems the threat model of explicitly flipping pixels or otherwise changing the inputs directly is less relevant than the case of identifying bad inputs in the environments (which the authors rename "natural observations") that cause the controller under attack to leave its stable region and end up in bad states for itself, unable to recover sometimes. This paper is an empirical exploration of this phenomenon, using as examples trained agents in a Gym environment and involving two humanoid robot simulations interacting with each other in various ways. The original policies are taken from previous training, so the main experimental contribution here is to solve an RL problem on the attacker's side to try to find the bad regions in the victim's control space, going via the means of the observation-based coupling.

I think these are timely issues. I agree that, to the extent that DRL policies are being seriously considered as candidates for various practical problems, we should seriously ask questions about the (lack of) stability of solutions.

I also think that the paper would be stronger if the authors considered the rich history in control design, formal methods and other communities where this approach to counter-example based thinking is fairly standard and in particular where the use of optimization methods to find 'bugs' has been studied in some depth.

So, for instance, the authors give us useful quantification to support the observation that the process of attacker RL finds ways to put the victim outside its stable region where it is naturally unstable and ineffective. However, they do not observe that there is a literature on systematically performing 'controller testing' via optimization using various techniques. Here are just a few examples to show the diversity:
Ghosh, S., Berkenkamp, F., Ranade, G., Qadeer, S., & Kapoor, A. (2018, May). Verifying controllers against adversarial examples with Bayesian optimization. In 2018 IEEE International Conference on Robotics and Automation (ICRA) (pp. 7306-7313). IEEE.
Ravanbakhsh, H., & Sankaranarayanan, S. (2016, October). Robust controller synthesis of switched systems using counterexample guided framework. In 2016 international conference on embedded software (EMSOFT) (pp. 1-10). IEEE.

The authors might find it interesting to note that this approach of viewing the control problem 'backwards' to generate new instances has a deeper history in control theory (of course without discussion of NNs etc), e.g.,
Doyle, J., Primbs, J. A., Shapiro, B., & Nevistic, V. (1996, December). Nonlinear games: examples and counterexamples. In Proceedings of 35th IEEE Conference on Decision and Control (Vol. 4, pp. 3915-3920). IEEE.

All that said, I find the results presented here to be plausible and pointing in the desired direction for exploring how to robustify DRL. The authors defer training using these examples to future work but I think the paper would be more self-contained if that were actually demonstrated here already. For reasons I mention above, the existence of this kind of weakness in controllers is not really surprising to people who have thought deeply about control, and it is only to be expected that NN based parameterisations of control would only be even more vulnerable. So, the more satisfying result would be the positive one that shows how to train DRL to be (more) robust under such attacks.



**Experience Assessment:**

I have published one or two papers in this area.

**Review Assessment: Checking Correctness Of Derivations And Theory:**

I carefully checked the derivations and theory.

**Review Assessment: Checking Correctness Of Experiments:**

I assessed the sensibility of the experiments.

**Review Assessment: Thoroughness In Paper Reading:**

I read the paper thoroughly.

---

> ### Author Response · Authors · 2019-11-09
> **Response**
>
> Thank you for your thoughtful review and the references to related work. We were unaware of this literature but agree it is highly relevant. We have revised the Related Work section in our submission to discuss the papers you mentioned and several others we discovered while reviewing the literature. We would appreciate any further feedback you may have on our revised submission.
>
> You also highlight the need for training Deep RL to be more robust to these attacks. We wholeheartedly agree: we deferred this to future work only because we felt there was insufficient space to adequately explore defenses in this paper. However, after feedback from yourself and other reviewers we agree it is worth including some simple baseline defenses in this paper. We have started work on a defence based on adversarial training, and will revise the paper early next week after further investigating our results.
>
> Our preliminary results suggest that the efficacy of the adversarial policy can be reduced by finetuning the victim on a mixture of episodes against the adversary and a normal opponent. (Finetuning against just the adversary works even better, but the victim catastrophically forgets how to play against a normal opponent.) Unfortunately, the victim is hardened only against the original adversary: retraining the adversary recovers comparable performance to the original victim.
>
> We also wanted to clarify that we do not consider the existence of weaknesses in deep RL policies to be in itself surprising. Indeed, any policy not in a global Nash equilibria can be exploited by some worst-case opponent. The result we found striking was that the worst-case opponent is a policy that is manifestly less capable than normal opponents which the victim plays successfully against. This suggests that whatever feature representation the deep RL policy has learnt is quite different from that of human sports players. Apart from being of interest scientifically, we believe this is important for practitioners to be aware of: testing an RL policy against opponents that seem challenging to a human may give a false assurance, as the worst-case opponent may in fact be a policy that humans would dismiss as being of little to no interest.

---

> > ### Comment · AnonReviewer2 · 2019-11-11
> > **Valid point**
> >
> > I agree with your observation:
> > "We also wanted to clarify that we do not consider the existence of weaknesses in deep RL policies to be in itself surprising. Indeed, any policy not in a global Nash equilibria can be exploited by some worst-case opponent. The result we found striking was that the worst-case opponent is a policy that is manifestly less capable than normal opponents which the victim plays successfully against. This suggests that whatever feature representation the deep RL policy has learnt is quite different from that of human sports players. Apart from being of interest scientifically, we believe this is important for practitioners to be aware of: testing an RL policy against opponents that seem challenging to a human may give a false assurance, as the worst-case opponent may in fact be a policy that humans would dismiss as being of little to no interest."

---

> ### Author Response · Authors · 2019-11-14
> **Defence Update**
>
> We have added Section 6 to the paper that demonstrates how to use adversarial policies to train a more robust victim policy. We would welcome any additional feedback on defences. Videos are still being generated, we will upload on the website soon.

---

> > ### Author Response · Authors · 2019-11-15
> > **Defence Update v2**
> >
> > We have made our final revision of the paper, updating the intro & related work in light of the defences. We have also updated the website to include videos of the fine-tuned victims and new adversaries, they are accessible under Load Preset->"You Shall Not Pass - Defences" and "You Shall Not Pass - Retrained Adversary".

---

### Decision · Program_Chairs · 2019-12-19

**Decision:**

Accept (Poster)

**Comment:**

This paper demonstrates that for deep RL problems one can construct adversarial examples where the examples don't really need to be even better than the best opponent. Surprisingly, sometimes, the adversarial opponent is less capable than normal opponents which the victim plays successfully against, yet they can disrupt the policies. The authors present a physically realistic threat model and demonstrate that  adversarial policies can exist in this threat
model.

The reviewers agree with this paper presents results (proof of concept) that is "timely" and the RL community will benefit from this result. Based on reviewers comment, I recommend to accept this paper.